

# Validation of low-density lipoprotein cholesterol equations in pediatric population

Gözde Ertürk Zararsız[1,2], Serkan Bolat[3], Ahu Cephe[4], Necla Kochan[5], Serra Ilayda Yerlitaş[1,2], Halef Okan Doğan[3] and Gökmen Zararsız[1,2]

[1] Department of Biostatistics, Erciyes University, Kayseri, Turkey
[2] Drug Application and Research Center (ERFARMA), Erciyes University, Kayseri, Turkey
[3] Department of Biochemistry, Cumhuriyet University, Sivas, Turkey
[4] Rectorate, Erciyes University, Kayseri, Turkey
[5] Izmir Biomedicine and Genome Center, Izmir, Turkey

Corresponding author
Gözde Ertürk Zararsız,
gozdeerturk9@gmail.com

## ABSTRACT

Several studies have shown a high prevalence of dyslipidemia in children. Since childhood lipid concentrations continue into adulthood, recognition of lipid abnormalities in the early period is crucial to prevent the development of future coronary heart disease (CHD). Low density lipoprotein cholesterol (LDL-C) is one of the most used parameters in the initiation and follow-up of treatment in patients with dyslipidemia. It is a well known fact that LDL-C lowering therapy reduces the risk of future CHD. Therefore, accurate determination of the LDL-C levels is so important for the management of lipid abnormalities. This study aimed to validate different LDL-C estimating equations in the Turkish population, composed of children and adolescents. A total of 3,908 children below 18 years old at Sivas Cumhuriyet University Hospital (Sivas, Turkey) were included in this study. LDL-C was directly measured by direct homogeneous assays, *i.e.*, Roche, Beckman, Siemens and estimated by Friedewald's, Martin/Hopkins', extended Martin-Hopkins' and Sampson's formulas. The concordances between the estimations obtained by the formulas and the direct measurements were evaluated both overall and separately for the LDL-C, triglycerides (TG) and non-high-density lipoprotein cholesterol (non-HDL-C) sublevels. Linear regression analysis was performed and residual error plots were generated between each estimation and direct measurement method. Coefficient of determination ($R^2$) and mean absolute deviations were also evaluated. The overall concordance of Friedewald, Sampson, Martin-Hopkins and the extended Martin-Hopkins formula were 64.6%, 69.9%, 69.4%, and 84.3% for the Roche direct assay, 69.8%, 71.6%, 73.6% and 80.4% for the Siemens direct assay, 66.5%, 68.8%, 68.9% and 82.1% for the Beckman direct assay, respectively. The extended Martin-Hopkins formula had the highest concordance coefficient in both overall and all sublevels of LDL-C, non-HDL-C, and TG. When estimating the LDL-C categories, the highest underestimation degrees were obtained with the Friedewald formula. Our analysis, conducted in a large pediatric population, showed that the extended Martin-Hopkins equation gives more reliable results in estimation of LDL-C compared to other equations.

## INTRODUCTION

Cardiovascular disease (CVD) has been found to be the leading cause of morbidity and mortality in children and adolescents worldwide, as well as adults. Dyslipidemia is a condition related to the abnormal lipoprotein metabolism and a common CVD risk factor. It causes atherosclerotic lesions in children aged 2–16 which can lead to an increased risk of cardiovascular disease in their adulthood (*Berenson et al., 1998*; *Daniels, Greer & on Nutrition, 2008*). It is an important clinical problem in children and adolescents especially living in Western countries due to the dietary causes and increasing incidence of obesity. According to the NHANES report obesity affects 18.5 percent of all American youth and 20.6 percent of adolescents (*Sanyaolu et al., 2019*). Although CVD does not usually manifest itself until the fourth decade of life, atherosclerosis, which is related to dyslipidemia, is known to begin at earlier stages of childhood. In fact, it has been shown that the early symptoms/signs of atherosclerosis, such as fatty lines, can appear as early as age 2 (*Berenson et al., 1998*). Hence, if the signs of the development of CVD can be diagnosed in childhood, cardiovascular disease can be possibly prevented or at least kept under control in adults by an earlier intervention such as the administration of lowering lipid levels or controlling lipid profiles which have been found to be helpful for primary and secondary prevention of the disease (*McGill Jr et al., 2001*; *Gidding, 2001*; *Fox et al., 2016*).

Low-density lipoprotein cholesterol (LDL-C), one of the most atherogenic lipoproteins and one of the critical parameters used for CVD risk assessment (*American Academy of Pediatrics, 2011*), has been identified as the key target for cholesterol-lowering therapy to minimize the risk of CVD (*Garoufi et al., 2017*). Therefore, it is important to be able to accurately measure the LDL-C serum levels to determine the further treatment of children and adolescents. Beta-quantification is regarded to be the gold standard method for LDL-C measurement; however, it has a number of drawbacks that limit its use in many laboratories due to its high cost and the test being time-consuming. For this reason, other direct methods, as well as the equations developed for LDL-C estimation, have been used to predict the LDL-C serum levels. Friedewald is one of those equations, commonly used for estimating LDL-C serum concentrations (*Friedewald, Levy & Fredrickson, 1972*). However, it has some limitations in practice. The main experimental limitation is the lack of ultracentrifugation measurement and reliance on direct homogeneous assays, which is transparently reported. Apart from that, it overestimates LDL-C levels when triglycerides (TG) are very low and underestimates LDL-C levels when TG levels are between 150 and 399 mg/dL (*Martin et al., 2013a*) or higher than 400 mg/dL. Also, it requires eight hours of overnight fasting which can be challenging for children and adolescents.

Due to the limitations of the Friedewald equation, other formulas have been developed such as Martin/Hopkins, Sampson, etc. in order to estimate LDL-C levels with higher accuracy (*Harry et al., 2000*; *Martin et al., 2013b*; *Palmer et al., 2019*; *Sampson et al., 2020*).

Presently, there has been limited testing of these formulas resulting in a gap in the literature for estimating LDL-C levels in children and adolescents (*Garoufi et al., 2017*; *Roper et al., 2017*; *Cicero et al., 2021*; *Molavi et al., 2020*).

Therefore, in this study, we aimed to investigate the validity of the LDL-C levels estimated by Friedewald, Martin-Hopkins, extended Martin-Hopkins and Sampson formulas with the LDL-C levels measured by some direct assays (*i.e.*, Roche, Beckman and Siemens) using a large cohort of Turkish children and adolescents.

## MATERIALS AND METHODS

### Study population

The study was carried out in a retrospective design. The study includes 3.908 children and adolescents under the age of 18 who were referred to the Department of Biochemistry from different units of Sivas Cumhuriyet University Medical Hospital and whose lipid measurements (HDL-C, LDL-C, TG and TC) were analyzed between March 3, 2011 and December 31, 2019. The study was conducted in accordance with the Declaration of Helsinki and adhered to Good Clinical Practice guidelines. The study was approved by the Ethics Committee of Sivas Cumhuriyet University (2022-03/17). Written informed consent was obtained from all individuals included in this study.

### Lipid measurements

Roche Cobas 8000, c-702 and c-501, Siemens Advia 1800, and Beckman Coulter AU5800 systems were used for direct measurement of HDL-C, LDL-C, TG and TC. In Siemens direct assay, HDL-C levels were measured with Trinder reaction. All other measurements in this and other direct assays are measured with colorimetric enzymatic reaction.

### Lipid estimations

To estimate LDL-C concentration, we used Friedewald (*Friedewald, Levy & Fredrickson, 1972*), Sampson (*Sampson et al., 2020*), Martin/Hopkins and extended Martin/Hopkins formulas (*Martin et al., 2013b*). All formulas and additional information can be found in the related papers (*Friedewald, Levy & Fredrickson, 1972*; *Martin et al., 2013b*; *Sampson et al., 2020*; *Zararsız et al., 2022*). We note here that, in the extended Martin-Hopkins LDL-C estimation formula (LDL-$C^E$), strata specific median TG/VLDL-C ratios were calculated using Turkish pediatric population data.

### Statistical analysis

The consistencies of each LDL-C estimating equation and direct assay were calculated using the overall concordance statistic. This statistic was defined as the ratio of direct LDL-C (LDL-$C^D$) in the same category as estimated LDL-C based on estimated LDL-C levels (< 110 mg/dL, 110 to 129 mg/dL, and ≥ 130 mg/dL). The concordance statistic was also calculated for TG and non-HDL-C sublevels (TG sublevels: < 75 mg/dL, 75 to 129 mg/dL, and ≥ 130 mg/dL; non-HDL-C sublevels: < 120 mg/dL, 120 to 144 mg/dL, and ≥ 145 mg/dL). To compare the estimated and measured LDL-C levels, ordinary least squares regression analysis models were fitted for each equation and each direct assay.

For each of this combination, residual error plots were generated to display the change of the difference between measured and estimated LDL-C levels according to TG levels. Moreover, precision–recall analyses were applied separately for each assay, for participants with LDL-C level < 110 mg/dL and LDL-C level ≥ 130 mg/dL. All statistical analyses were performed using R 4.0.4 (www.r-project.org) statistical software.

## RESULTS

### Patient characteristics

Table 1 shows the demographic characteristics of all participants. A total of 3,908 children were included in this study, 54.7% of which were female and 45.3% were male. The mean age of the children who took part in this study was $12.1 \pm 4.6$. The median of LDL-C levels measured by direct method, TC levels, TG levels, and HDL-C levels were 92 mg/dL, 153 mg/dL, 99 mg/dL, and 44.3 mg/dL, respectively. The median nonHDL-C level was 107 mg/dL, and the TG/TC ratio was 0.66 mg/dL. Out of the 3,908 lipid profiles considered in this study, 2,356 (60.3%) were measured with Roche, 893 (22.9%) with Beckman, and 659 (16.9%) with Siemens direct assays. The demographic characteristics of each participant were also provided separately for each assay (Table 1).

### Comparison of LDL-C concentrations calculated by various formulas *versus* direct assays
#### *Overall concordances of the different equations for LDL-C estimation*

Strata-specific median ratios of TG/VLDL-C were used to estimate LDL-C levels. These estimations were made using the extended Martin-Hopkins' formula. In order to see how the estimates change at different TG and non-HDL-C levels, the results of the calculations made according to the cut-off values determined for these variables are shown in Table S1. This two-dimensional cross-table contained the median ratio of TG/VLDL-C with 180 cells were generated, with TG sublevels in the rows and non-HDL-C sublevels in the columns. The cells, in this 180-cell table, display the median statistics for TG/VLDL-C ratio.

Overall concordances of LDL-C estimates for each assay are given in Fig. 1. It can be seen that the extended Martin-Hopkins formula produced the highest and Friedewald formula produced the lowest concordances within each direct assay. In Siemens direct assay, the concordance of Martin-Hopkins' formula was higher than the Sampson's formula. In Roche and Beckman direct assays, the performances of these two formulas were found to be very similar.

### The distribution density of LDL-C concentrations calculated by direct methods and different formulas

In Fig. 2, the raincloud plots show the distributions of the LDL-C levels measured by direct assays and estimated by Friedewald, Sampson, Martin-Hopkins and extended Martin-Hopkins equations. The red line in this figure displays the difference between the median of estimated LDL-C levels by each equation and the median of LDL-C levels measured by direct methods. The figure shows that, with the exception of the extended Martin-Hopkins' formula, all formulas underestimated LDL-C levels when Roche and

**Table 1** Study population characteristics.

| Characteristic | Overall ($N = 3,908$) | Roche ($N = 2,356$) | Beckman ($N = 893$) | Siemens ($N = 659$) |
|---|---|---|---|---|
| Age (years) | $12.1 \pm 4.6$ | $12.1 \pm 4.3$ | $12.4 \pm 4.8$ | $11.7 \pm 5.2$ |
| Gender | | | | |
| Female | 2,138 (54.7) | 1,363 (57.9) | 451 (50.5) | 324 (49.2) |
| Male | 1,770 (45.3) | 993 (42.1) | 442 (49.5) | 335 (50.8) |
| Lipid values | | | | |
| TC (mg/dL) | 153.0 (131.0–176.0) | 150.0 (129.0–171.8) | 164.0 (141.0–187.5) | 149.0 (127.0–174.0) |
| TG (mg/dL) | 99.0 (72.0–140.0) | 103.0 (74.0–143.0) | 93.0 (68.0–129.0) | 99.0 (70.0–145.0) |
| HDL-C (mg/dL) | 44.3 (37.5–52.0) | 45.0 (38.0–52.0) | 46.0 (39.0–53.0) | 42.1 (35.3–50.0) |
| Non-HDL-C (mg/dL) | 107.0 (87.0–129.0) | 103.0 (85.0–126.0) | 117.0 (98.0–139.0) | 106.0 (86.0–128.7) |
| TG - TC ratio | 0.7 (0.5–0.9) | 0.7 (0.5–1.0) | 0.6 (0.4–0.8) | 0.7 (0.5–0.9) |
| LDL-C$^D$ (mg/dL) | 92.0 (74.0–113.0) | 91.0 (73.0–112.0) | 104.0 (86.0–121.0) | 81.0 (64.0–101.0) |
| LDL-C$^F$ (mg/dL) | 84.4 (66.4–104.8) | 80.0 (63.4–100.0) | 96.4 (77.8–116.5) | 84.2 (64.1–103.1) |
| LDL-C$^S$ (mg/dL) | 86.4 (67.8–107.0) | 82.1 (65.1–102.5) | 97.5 (79.7–118.2) | 86.0 (66.1–106.3) |
| LDL-C$^M$ (mg/dL) | 86.4 (68.9–106.6) | 82.6 (65.9–102.7) | 96.9 (79.7–116.7) | 85.7 (67.9–106.2) |
| LDL-C$^E$ (mg/dL) | 93.3 (76.9–114.1) | 91.7 (75.1–112.4) | 104.6 (89.1–121.0) | 83.2 (67.2–101.2) |

**Notes.**

Values are expressed as $N(\%)$, mean $\pm$ SD or median (1st–3rd quartiles).

TC, total cholesterol; TG, triglycerides; HDL-C, high-density lipoprotein cholesterol; LDL-C, low-density lipoprotein cholesterol; Non-HDL-C, non-high-density lipoprotein cholesterol; LDL-C$^D$, LDL-C measured by direct assay; LDL-C$^F$, LDL-C calculated by Friedewald formula; LDL-C$^S$, LDL-C calculated by Sampson formula; LDL-C$^M$, LDL-C calculated by Martin-Hopkins formula; LDL-C$^E$, LDL-C calculated by the extended Martin-Hopkins formula.

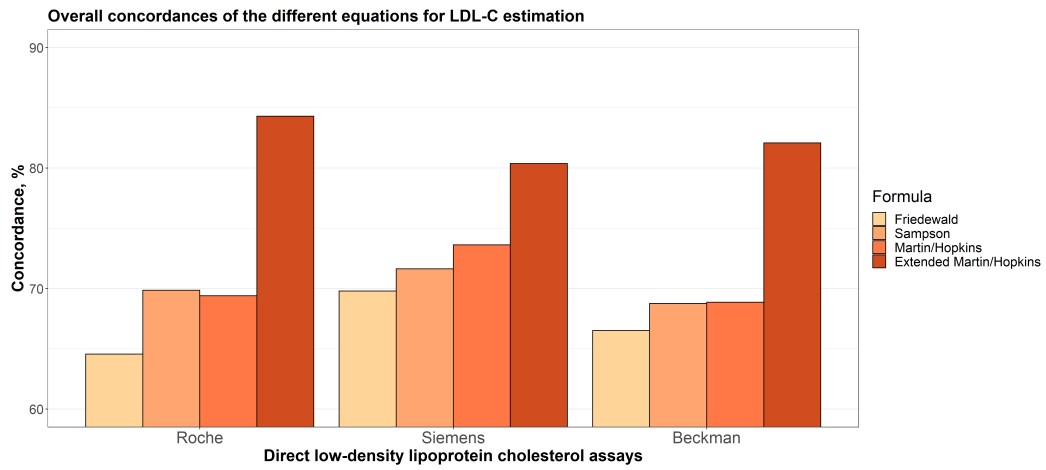

**Figure 1** Overall concordances of the different equations for LDL-C estimation.

Beckman direct assays were employed, but overestimated LDL-C levels when Siemens direct assay was performed. The distribution pattern for the Siemens assay, however, was the most similar with Martin-Hopkins' formula.

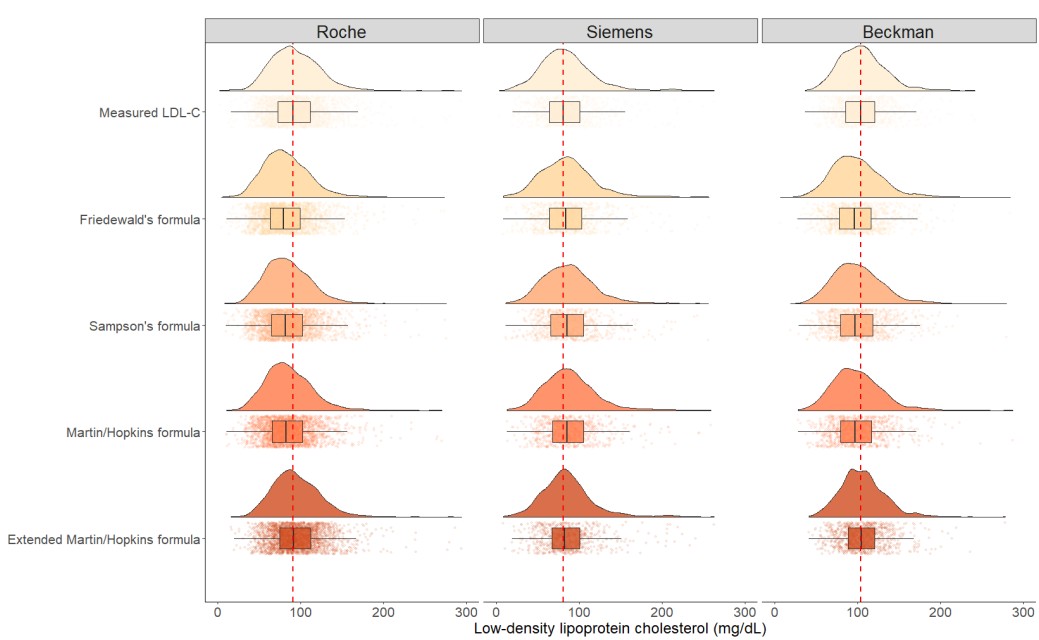

**Figure 2** The distribution density of LDL-C concentrations evaluated by direct methods and different formulas.

## Concordances of the different equations for LDL-C estimation by LDL-C strata

In Fig. 3, the concordances of the different equations for LDL-C estimation by different LDL-C sublevels are given for each assay. In all cases, the extended Martin-Hopkins equation gave the most concordant results. When LDL-C is less than 110 mg/dL, Friedewald equation gave the least concordant results for each assay. The Sampson formula performed slightly better than the Martin-Hopkins equation for the Roche direct assay, but slightly worse than the Martin-Hopkins equation for the Siemens direct assay. Very similar results were observed for these two equations for the Beckman assay. When LDL-C levels were between 110 and 129 mg/dL, again, lowest concordances were obtained with the Friedewald formula. The performance of the Martin-Hopkins equation was higher than Sampson equation for Roche and Beckman direct assays. These two approaches have approximately equal performance for Siemens direct assay. When LDL-C levels were higher than 130 mg/dL, the performance of the Sampson formula was relatively higher than the Martin-Hopkins equation for all direct assays. The performance of the Friedewald equation was higher than Sampson and the Martin-Hopkins equation for Siemens direct assay.

seclevel2

## Concordances of the different equations for LDL-C estimation by triglycerides strata

Overall concordances for LDL-C estimates by TG sublevels (< 75 mg/dL, 75 to 129 mg/dL and ≥ 130 mg/dL) were given for each assay Fig. S1. The results showed that the extended

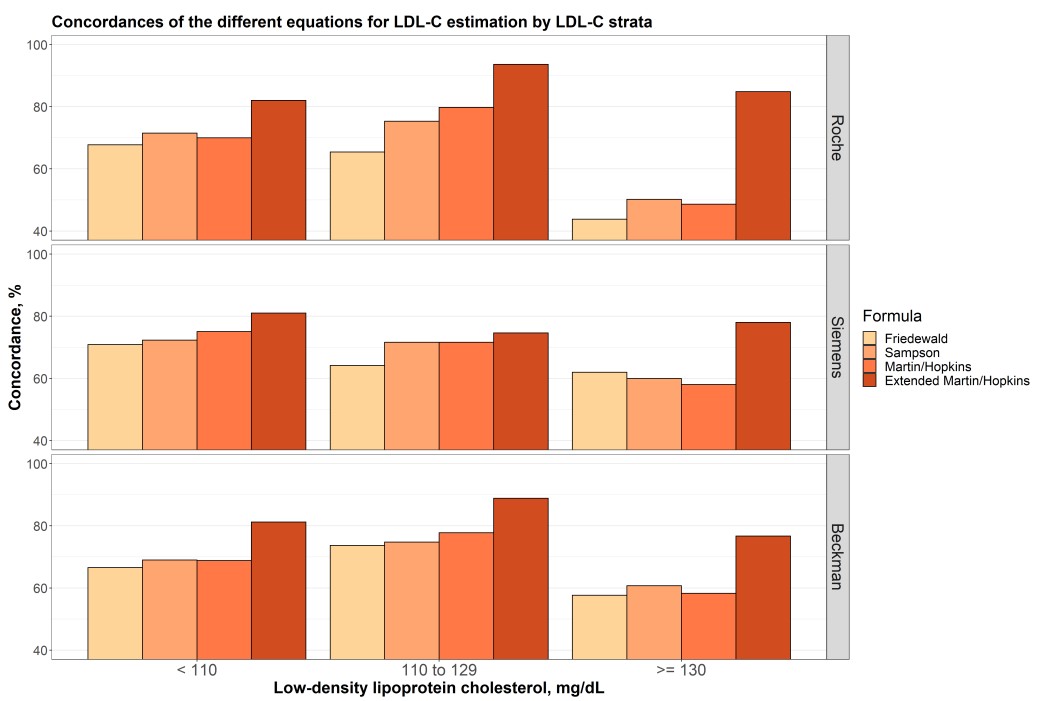

**Figure 3** **Concordances of the different equations for LDL-C estimation by LDL-C strata.**

Martin-Hopkins gave the highest concordance for each assay in any TG sublevels. However, as the TG levels increased, the concordances of the equations decreased for any direct assay. The concordance of the equations used for LDL-C estimation given by TG sublevels with different LDL-C strata was also calculated (Figs. S2–S4).

When the performances of the methods were evaluated according to the TG changes in LDL-C sublevels, we found that the results differed from assay to assay (Figs. S2–S4). In patients with LDL-C < 110 mg/dL, the extended Martin-Hopkins equation gave the highest performance by comparison with other equations, *i.e.,* Friedewald, Sampson and Martin-Hopkins for all TG levels. Remarkably, the extended Martin-Hopkins equation decreased slightly for Roche and Beckman direct assays, whereas this formula had an almost equal performance for Siemens direct assay as the TG level increased from < 75 mg/dL to ≥ 130 mg/dL. When the TG was < 75 or between 75 mg/dL and 129 mg/dL, Friedewald and Sampson equations had a similar performance for each assay. However, the performance of Friedewald method was lower than Sampson method when TG level was ≥ 130 mg/dL for each assay. Even though the performance of the Martin-Hopkins equation decreased as the TG level increased for Roche and Beckman direct assays, this formula had an almost equal performance for Siemens direct assay. The performance of Friedewald formula significantly reduced as the TG level increased for Roche and Beckman direct assay. For LDL-C between 110 to 129 mg/dL, the performance of the extended Martin-Hopkins equation was again the highest for each assay when TG level was < 75 mg/dL. The concordance of the extended Martin-Hopkins approach decreased as TG level

increased for Roche direct assay, whereas firstly, it decreased then, increased for Beckman and Siemens direct assays. In TG < 75 mg/dL, as Friedewald and Sampson had a similar performance for each assay, the distinction between these equations increased while the TG level increased for each assay. Even though the performance of Friedewald, Sampson and Martin-Hopkins equations generally decreased while the TG level increased for Roche and Beckman direct assays, these performances first decreased and then increased for Siemens direct assay. In patients with LDL-C ≥ 130 mg/dL, the performance of the Martin-Hopkins and the extended Martin-Hopkins equations constantly decreased while the TG levels were increasing for Roche direct assay; however, it first increased and then decreased for Siemens and Beckman direct assay. Friedewald and Sampson formulas showed the same performance in the TG level < 75 mg/dL for each assay. However, while this situation changed in the TG levels between 75 and 129 mg/dL for Roche and Siemens direct assays, it remained the same for Beckman direct assay. While the performance of the Friedewald equation was lower than Sampson equation in the TG levels between 75 and 129 mg/dL for Roche direct assay, it had a reverse situation for Siemens direct assay.

While the performance of the Friedewald equation was lower than Sampson equation for Roche and Beckman direct assays, these performances were the same for Siemens direct assay at the TG level ≥ 130 mg/dL.

## Concordances of the different equations for LDL-C estimation by non-HDL-C strata

Figure S5 displays the overall concordances for LDL-C estimates by non-HDL-C sublevels (< 120 mg/dL, 120 to 144 mg/dL and ≥ 145 mg/dL). The extended Martin-Hopkins method produced the highest concordance for each assay and all non-HDL-C sublevels. Furthermore, for Roche and Siemens direct assays, the concordance of the Friedewald method decreased as non-HDL-C levels increased. The concordance of the methods used to estimate LDL-C levels given by non-HDL-C sublevels with different LDL-C strata was also given (Figs. S6–S8).

When the performances of the methods were evaluated according to the non-HDL-C changes in LDL-C sublevels, we found that the results differed from assay to assay (Figs. S4–S6). In patients with LDL-C < 110 mg/dL, for each assay, while the non-HDL-C levels increased, the concordance of the extended Martin-Hopkins methods decreased. The performance of Friedewald, Sampson and the Martin-Hopkins equations were almost similar in the non-HDL-C level < 144 mg/dL; however, this situation changed when non-HDL-C exceeded 144 mg/dL. For LDL-C between 110 to 129 mg/dL, while the non-HDL-C levels increased, the concordance of the Martin-Hopkins increased, but the concordance of the extended Martin-Hopkins decreased for Roche direct assay. In patients with LDL-C ≥ 130 mg/dL, for each assay and all sublevels of the non-HDL except non-HDL-C was between 120 mg/dL and 144 mg/dL for Siemens direct assay, the extended Martin-Hopkins gave the highest concordance.
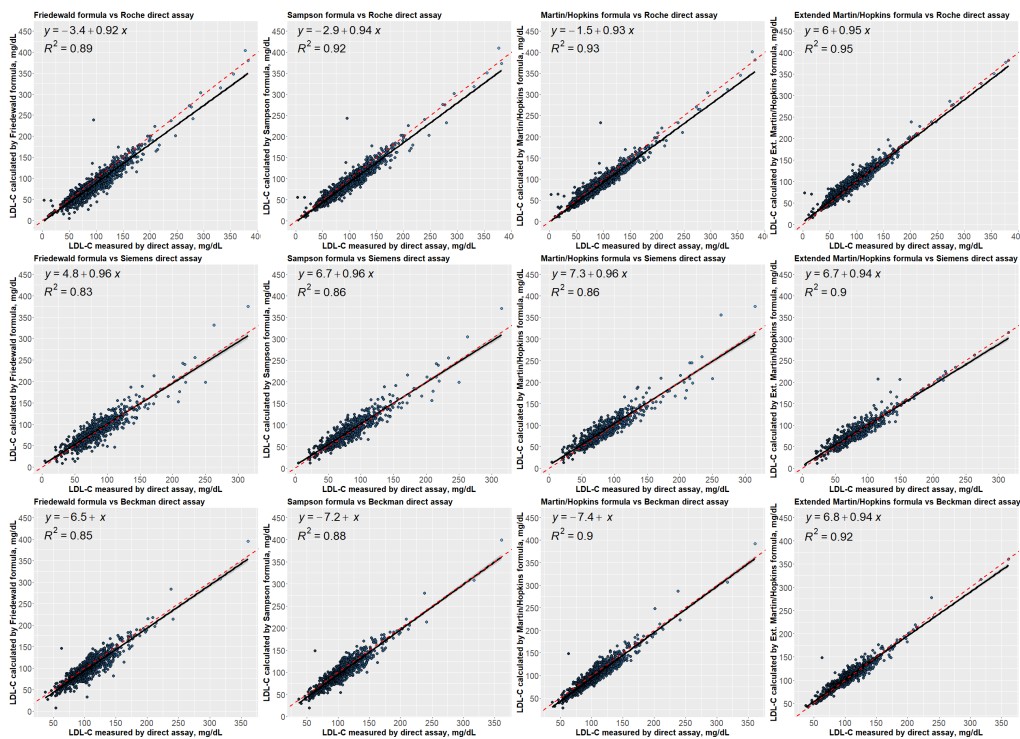

**Figure 4** Regression analysis between LDL-C levels estimated by formulas and directly measured LDL-C levels.

## Regression analysis between LDL-C levels estimated by formulas and directly measured LDL-C levels

In Fig. 4, the linear regression analyses were performed to examine the correlation between the LDL-C levels estimated by formulas and measured by direct assays. These results revealed that estimated LDL-C levels by extended Martin-Hopkins formula indicate a better correlation with each assay. The highest R square statistic was obtained with the extended Martin-Hopkins methods with an R square of 0.95 for Roche direct measurement method, an R square of 0.90 for Beckman direct measurement method and an R square of 0.92 for Siemens direct measurement method. It is obvious to see that the extended Martin-Hopkins shows a better association with any direct methods overall.

## Residual error plots for LDL-C by different formulas with respect to different direct assay methods

The residual error plots are given in Fig. S9. These plots demonstrate how the difference between LDL-C estimates calculated by the equations and directly measured LDL-C levels varies with triglyceride levels. For each assay, the Friedewald and Sampson formula was found to underestimate LDL-C levels when TG levels were elevated. A similar pattern was observed in the Martin-Hopkins formula for the Roche and Beckman direct assays. According to TG levels, the difference between LDL-C estimates by Martin-Hopkins formula and Siemens direct assay was less compared to other assays. The difference in each

assay for the Extended Martin-Hopkins formula was found to be nearly zero and did not change with TG levels. The extended Martin-Hopkins formula produced the lowest mean absolute deviation statistics for each assay.

### The proportion of misclassified samples per direction by estimated low-density lipoprotein cholesterol (LDL-C) category

In Fig. S10, diverging bar charts show the total percentage of samples who were underclassified and overclassified within each LDL-C category. While drawing these graphs, LDL-C is considered to be divided into six categories ($< 70$ mg/dL, 70 to 99 mg/dL, 100 to 129 mg/dL, 130 to 159 mg/dL, 160 to 189 mg/dL and $\geq 190$ mg/dL). In each bar, how well different equations concordant these six categories or how many categories they predict up/down are expressed in different colors. The concordance of the extended Martin-Hopkins equations was the highest according to other equations in all sublevels of the LDL-C levels and each assay. In LDL-C levels between 110 to 129 mg/dL, the underestimation of LDL-C occurred in 0.7% with the extended Martin-Hopkins equation compared with 34.1% with the Friedewald equation, 24.1% with the Sampson equation and 20.0% with the Martin-Hopkins equation for Roche direct assay.

In LDL-C $< 110$ mg/dL, the degree of underestimation was least pronounced with the extended Martin-Hopkins equation, with only 7.6% of patients underclassified for Roche assay and with only 9.0% of patients underclassified for Beckman assay. In Roche and Beckman direct assays, this statistic was 31.6% and 29.3% for the Friedewald equation, 27.3% and 29.3% for the Sampson equation, 27.7% and 27.4% for the Martin-Hopkins equation, respectively. In general, the concordant estimation of LDL-C for the Friedewald equation was the lowest according to other equations in all sublevels of the LDL-C levels and each assay. In most cases, LDL-C levels were overestimated in Sampson and the Martin-Hopkins equations. For instance, in LDL-C levels between 110 and 129 mg/dL, the degree of overestimation with Sampson and the Martin-Hopkins formulas were 20.9%, while this degree was 19.4% for Friedewald and 10.5% for the extended Martin-Hopkins in Siemens direct assay. The highest degree of underestimation was observed in the Friedewald equation for all subclasses of LDL-C levels and all types of assays.

Precision–recall curves are given for each LDL-C estimating equation for each assay separately in Fig. 5. In participants with LDL-C $< 110$ mg/dL, Martin-Hopkins and extended Martin-Hopkins equations had the highest area under the curve (AUC) statistics for each assay. In participants with LDL-C $\geq 130$ mg/dL, extended Martin-Hopkins formula was the overall winner. The lowest AUC statistics were obtained for Friedewald equation in all scenarios. These precision–recall analyses supported the results of Fig. S10. In summary, the degree of underestimation and overestimation was the lowest with the extended Martin-Hopkins equation.

## DISCUSSION

The disadvantages of the Friedewald formula, such as requiring fasting serum and underestimating LDL-C levels below 70 and above 150 mg/dL, led to the recommendation of new formulas. Martin-Hopkins and Sampson are among these formulas and validation

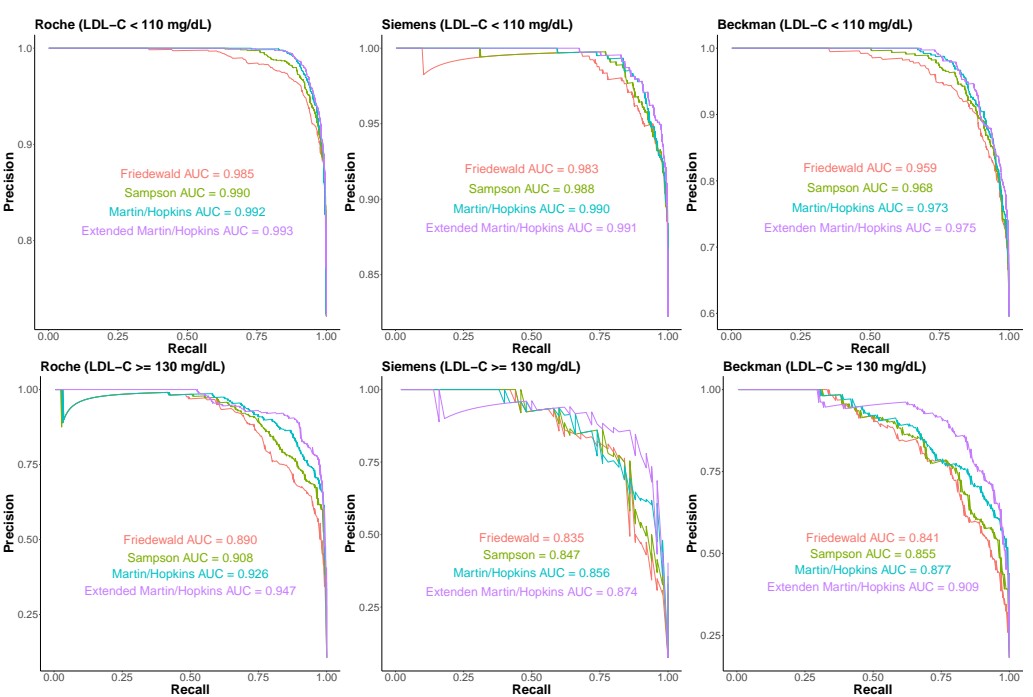

**Figure 5** The proportion of misclassified samples per direction by estimated low-density lipoprotein cholesterol (LDL-C) category.

of these formulas were examined for adult data in several populations (*Chaen et al., 2016*; *Sathiyakumar et al., 2018*; *Shin, Bohra & Kongpakpaisarn, 2018*; *Sathiyakumar, Blumenthal & Elshazly, 2020*; *Zafrir, Saliba & Flugelman, 2020*; *Song et al., 2021*; *Piani et al., 2021*; *Vargas-Vázquez et al., 2021*; *Zararsız et al., 2022*). Studies investigating the validation of these recent formulations in the pediatric population are very limited. *Roper et al. (2017)* compared Friedewald and Martin-Hopkins formulas in a study of 127 pediatric data. The authors recommended the use of the Martin-Hopkins formula in cases where LDL-C is below 100 mg/dL and TG is between 150 and 399 mg/dL. *Garoufi et al. (2017)* investigated the Friedewald and Anandaraja formulas in the data of 1,005 children, 317 of which were dyslipidemic, and as a result suggested that the Friedewald formula may be a good screening tool, and the Anandaraja formula may be more appropriate in the follow-up of patients with dyslipidemia. *Cicero et al. (2021)* conducted a study with 2,605 pediatric data, compared Friedewald and Sampson formulas, and as a result stated that the Sampson formula was more reliable in estimating LDL-C. Our study, up till now, is the study with the largest sample size investigating the validation of LDL-C formulas in the pediatric population. Evaluation of the different direct assays in the performed comprehensive analyses, as well as the use of the extended Martin-Hopkins formula calculated from the median statistics of our own population are among the important contributions of our study. The results of our study revealed that there is a significant difference in the performance of the equations used in the LDL-C estimation for different assays. The method with the highest overall concordance coefficient was the extended Martin-Hopkins formula.
The extended Martin-Hopkins formula showed the best performance in all subgroups of LDL-C, non-HDL-C, and TG. The lowest performing equation was the Friedewald formula.

LDL-C cut-off points of < 110 mg/dL and ≥ 130 mg/dL have been used for the definition of acceptable and abnormal results in children and adolescence, respectively (*Lim et al., 2020*). In this study, we found higher concordance between the extended Martin-Hopkins formula and direct measurements compared to other formulas for both < 110 mg/dL and ≥ 130 mg/dL categories. Friedewald is the most commonly used formula in the clinical laboratory to estimate LDL-C values (*Vujovic et al., 2010*). Considering the category of ≥ 130 mg/dL in Roche direct assay, it was determined that the Friedewald formula underestimates 55% of the individuals compared to the direct measurement method. This ratio was found to be only 13.2% in the extended Martins-Hopkins formula. Similar results were obtained for Siemens and Beckman assays. Friedewald formula underestimates 26% and 38.7% of individuals, while the extended Martin-Hopkins formula underestimates 16% and 20.9% of individuals for Siemens and Beckman assays, respectively. Accordingly, we think that the use of Friedewald formula in children and adolescents may increase the possibility of false-negative diagnosis of dyslipidemia. Coronary artery disease (CAD) is the leading cause of global mortality and has a significant health-economic impact (*Gheorghe et al., 2018*; *Roth et al., 2020*). The number of CVD deaths gradually accelerated from 12.1 million to 18.6 million by 2019 (*Roth et al., 2020*). Increased LDL-C levels have been linked to an increased risk of CVD mortality (*Abdullah et al., 2018*). It has been shown that the average LDL-C reduction of 0.7 mmol/L (12.6%) led to a 19% risk reduction in the primary CVD death and nonfatal myocardial infarction. Moreover, regardless of age, sex, baseline LDL-C, or previous CVD status, a one mmol/L LDL-C reduction resulted in around 20% fewer CVD occurrences (*Stein & Raal, 2014*). Thus, we think that underestimation of the LDL-C levels may cause disruptions in the follow-up and treatment of patients and increase the risk of future CVD. Although the extended Martin-Hopkins has the highest concordance at both ≥ 110 mg/dL and 110-129 mg/dL categories with direct measurement, it was determined that the extended Martin-Hopkins formula overestimates 10.4% and 5.7% of individuals for Roche direct assay, 12% and 10.5% of individuals for Siemens direct assay, and 9.8% and 10.1% of individuals for Beckman direct assay respectively. In most cases, this percentage was found to be lower in the other formula for Roche and Beckman assays. Therefore, we think that the use of the extended Martins-Hopkins formula may increase the false positive rate, albeit at a low rate, thereby increasing the risk of unnecessary medical procedures. However, as a generalized result, it can be said that the extended Martin-Hopkins formula gave the lowest proportion of misclassified samples for all assays.

In case of high TG levels, chylomicrons accumulate at high levels and may change the association between TG and cholesterol. Therefore, Friedewald formula causes larger errors in LDL-C estimation (*Martin et al., 2013b*). Although many previous studies evaluated the effect of high TG levels on the accuracy of the equations developed for LDL-C estimation in adults, little is known regarding the effect of higher TG levels in the children and adolescence. The cut-off value of ≥ 130 mg/dL is being used for the definition of abnormal

high TG levels in children and adolescence. Thus, we evaluated the concordance between direct measurement and equations developed for LDL-C estimation at the TG levels $\geq$ 130 mg/dL. We determined that the extended Martin-Hopkins formula has higher concordance with direct measurement compared to other formulas. Accordingly, we think that the extended Martin-Hopkins formula may be used to obtain best performance in the estimation of the LDL-C levels in case of higher TG levels.

Our limitation in this study is that preparative ultracentrifugation methods or beta quantification was not used. Direct assays were used since they are cheaper and does not require highly manual technique. However, a major disadvantage of these assays is the lack of standardization. Our hypothesis of calculating median statistics from our own population should be evaluated with this limitation.

# CONCLUSION

The performance of the equations used in LDL-C estimation varies in different direct assays. The extended Martin-Hopkins formula may be a suitable equation when evaluating LDL-C levels in the pediatric population. When using Martin-Hopkins formula, calculating the population-based median statistics might be helpful to researchers to get more concordant results with the direct assays. Further validation is needed in different populations.

### Funding

This study was supported by the Research Fund of Erciyes University [TSG-2021-10912]. The funders had no role in study design, data collection and analysis, decision to publish, or preparation of the manuscript.

### Grant Disclosures

The following grant information was disclosed by the authors:
The Research Fund of Erciyes University: TSG-2021-10912.

### Competing Interests

The authors declare there are no competing interests.

### Author Contributions

- Gözde Ertürk Zararsız conceived and designed the experiments, performed the experiments, analyzed the data, prepared figures and/or tables, authored or reviewed drafts of the article, and approved the final draft.
- Serkan Bolat conceived and designed the experiments, prepared figures and/or tables, and approved the final draft.
- Ahu Cephe performed the experiments, analyzed the data, authored or reviewed drafts of the article, and approved the final draft.
- Necla Kochan performed the experiments, analyzed the data, authored or reviewed drafts of the article, and approved the final draft.
- Serra Ilayda Yerlitaş performed the experiments, analyzed the data, authored or reviewed drafts of the article, and approved the final draft.
- Halef Okan Doğan conceived and designed the experiments, authored or reviewed drafts of the article, and approved the final draft.
- Gökmen Zararsız conceived and designed the experiments, prepared figures and/or tables, authored or reviewed drafts of the article, and approved the final draft.

## Human Ethics

The following information was supplied relating to ethical approvals (i.e., approving body and any reference numbers):

The Ethics Committee of Sivas Cumhuriyet University approved the study (2022-03/17).

## Data Availability

The data cannot be shared publicly due to the restrictions of the Sivas Cumhuriyet University Ethics Committee.

The data are available upon a reasonable request for researchers who meet the criteria for access to confidential data. Data requests can be sent to a staff member in Sivas Cumhuriyet University, Faculty of Medicine and Department of Biochemistry: Demet Kablan, E-mail: demetekablan@gmail.com.

## Supplemental Information

Supplemental information for this article can be found online at http://dx.doi.org/10.7717/peerj.14544#supplemental-information.

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
