# Peer review of "Validation of low-density lipoprotein cholesterol equations in pediatric population"

_PeerJ, doi:10.7717/peerj.14544_

## Round 0.1 · original submission · Minor Revisions

Please see the comments from reviewers and revise accordingly

Reviewer 1 ·

Basic reporting

1. Clear, unambiguous and professional usage of English throughout the text.
2. The introduction and the body of the text contain ample references to relevant research, which makes it easy to understand related work.
3. Ample plots and tables are presented to help understand and explain the statistical basis of the authors’ conclusions.
4. Overall, a very well-written, self-contained and cross-referenced text with very clear logical flow.

Experimental design

1. Clear explanation and impact of research outlined. The objective is to compare concordances for 4 separate formulae for LDL-C estimation, when compared to experimental data from 3 independent assays.
2. The authors provide ample statistical tests for substantiating that the extended Martin-Hopkins formula is the most accurate estimate of LDL-C levels. Both the raw data and plots are provided, and I verified that they exactly agree.
3. Independent assays from 3 different sources (Roche, Siemens and Beckman) are used for the tests to minimize any errors from sample bias.
4. Overall, the experimental design is solid from a statistical perspective.

Validity of the findings

1. The raw data and plots from the statistical tests are provided, which agree with each other.
2. The statistical tests are thorough and exhaustive. Results from tests against 3 different assays agree and point towards the same conclusion that the extended Martin-Hopkins formula is the most accurate estimate of LDL-C levels.
3. R-squared-analysis and precision-recall tests are also presented, which agree with the stated concordances for the 4 LDL-C formulae under examination.

Additional comments

Suggest the authors to perhaps convert the data in Fig 8 into a precision-recall analysis. It is slightly hard to read the percentages in that figure.

Reviewer 2 ·

Basic reporting

Congratulations on putting together this comprehensive report.

I would suggest deleting the word "so" in two places in the abstract. Specifically:
-Line 16 change "early period is so crucial" to "early period is crucial"
-Line 20 change "accurate determination of the LDL-C levels is so important" to "accurate determination of the LDL-C levels is important"

On page 2, Lines 63-65, a revision in the description of the Friedewald equation is needed. Currently, it is written that "it overestimates LDL-C levels when triglycerides (TG) are very low and underestimates LDL-C levels when TG is higher than 400 mg/dL". It should also be noted that the Friedewald equation underestimates LDL-C levels when TG levels are 150-399 mg/dL, as shown in J Am Coll Cardiol 2013;62(8):732-9."https://pubmed.ncbi.nlm.nih.gov/23524048/

On page 13, line 321, the authors could delete "As we indicated in the previous paragraph,"

On page 12, line 274, the period is misplaced. Change "150 to 399 mg/dL Garoufi et al. (2017). to "150 to 399 mg/dL. Garoufi et al. (2017)" since the Garoufi is for the following sentence and the prior sentence relates to a different reference.

Experimental design

The main experimental limitation is the lack of ultracentrifugation measurement and reliance on direct homogeneous assays, which is transparently reported.

Validity of the findings

Page 13, lines 331-332 - Please consider softening the recommendation for researchers to calculate the extended Martin-Hopkins formula using median statistics of their own population. Although this is attractive to tailor the calculation to local population data, the challenge is that the quality of data may vary and this practice could create a lack of harmony in the determination of LDL-C values. It is not clear that at scale the practice of recalculating median values will be more accurate than using the original median values of the Martin-Hopkins formula, which are freely available and well validated. There is an advantage of labs around the world using a consistent approach based on strong evidence. Certain studies such as the FOURIER trial have validated the Martin approach in many countries around the world. It may be the case that the median values are consistent from population to population around the world once one accounts for TG and cholesterol levels, as is already embedded in the Martin formula. The Martin formula is validated against VAP ultracentrifugation and Beta quantification, whereas when recalculating median values, many labs would depend on direct homogeneous assays that are not well standardized and can introduce bias.

---

## Round 0.2 · Minor Revisions

I must applaud the authors for drafting an exciting manuscript. I also thank the author's response and for improvising the manuscript based on the reviewer's comments. The aim of the study is clear. However, as an editor, I have further concerns as below.

The aim of the study is clear. However, the manuscript needs many simplifications to improve the reader's comprehension.

1. Limit decimal points to 1. Table 1.
2. The triglyceride stratification is very granular. Suggest authors limit TG stratification to 6 to 8 levels (Current rows are 30+, recommend only 6-8 rows). Current Table 2, with 30+ rows, can be added as a supplement.
3. Figure 4. Concordance of the different equations for LDL-C estimation by triglycerides strata & Figure 5. Concordances of the different equations for LDL-C estimation by non-HDL-C strata.- would qualify for a supplement. However, the aim of the study did not include stratifying by TG. Therefore, to preserve the study's message's simplicity, figures 4 & 5 can be put in a supplement unless the authors strongly feel not to.
4. Figure 7. Residual error plots & Figure 9. The precision-recall curves are more of interest to a limited pool of readers with statistical backgrounds. Recommend this be included as a supplement.

---

## Round 0.3 · accepted · Accept

Thank you for making changes to the manuscript. Your manuscript is now accepted for publication. Congratulations!